# Wastewater-Based Epidemiology Mitigates COVID-19 Outbreaks at a Food Processing Facility near the Mexico-U.S. Border—November 2020–March 2022

**DOI:** 10.3390/v14122684

**Published:** 2022-11-30

**Authors:** Gabriel K. Innes, Bradley W. Schmitz, Paul E. Brierley, Juan Guzman, Sarah M. Prasek, Martha Ruedas, Ana Sanchez, Subhadeep Bhattacharjee, Stephanie Slinski

**Affiliations:** 1Yuma Center of Excellence for Desert Agriculture (YCEDA), University of Arizona, 6425 W. 8th St., Yuma, AZ 85364, USA; 2DatePac LLC, 2575 E 23rd Ln, Yuma, AZ 85365, USA; 3Water & Energy Sustainable Technology (WEST) Center, University of Arizona, 2959 W Calle Agua Nueva, Tucson, AZ 85745, USA

**Keywords:** wastewater-based epidemiology, SARS-CoV-2, surveillance, COVID-19, targeted clinical testing

## Abstract

**Background:** Wastewater-based epidemiology (WBE) has the potential to inform activities to contain infectious disease outbreaks in both the public and private sectors. Although WBE for SARS-CoV-2 has shown promise over short time intervals, no other groups have evaluated how a public-private partnership could influence disease spread through public health action over time. The aim of this study was to characterize and assess the application of WBE to inform public health response and contain COVID-19 infections in a food processing facility. **Methods:** Over the period November 2020–March 2022, wastewater in an Arizona food processing facility was monitored for the presence of SARS-CoV-2 using Real-Time Quantitative PCR. Upon positive detection, partners discussed public health intervention strategies, including infection control reinforcement, antigen testing, and vaccination. **Results:** SARS-CoV-2 RNA was detected on 18 of 205 days in which wastewater was sampled and analyzed (8.8%): seven during Wild-type predominance and 11 during Omicron-variant predominance. All detections triggered the reinforcement of infection control guidelines. In five of the 18 events, active antigen testing identified asymptomatic workers. **Conclusions:** These steps heightened awareness to refine infection control protocols and averted possible transmission events during periods where detection occurred. This public-private partnership has potentially decreased human illness and economic loss during the COVID-19 pandemic.

## 1. Introduction

The COVID-19 pandemic has rattled life’s normalcy, from personal and institutional health to the food and commodity supply chains [1]. Disruptions due to COVID-19 have led to challenges and strain throughout the food system [2], in large part due to worker illness and absenteeism [3]. Repercussions from COVID-19 illness in the food system workforce have manifested as increases in animal culling and food waste [4,5], in addition to food processor facility shutdowns [6], which manifest as significant food safety challenges and security challenges for consumers [7]. Therefore, minimizing SARS-CoV-2 infections among workers in the food sector is critical to preserving worker health and private industry operations [8].

Arizona is one of the largest agricultural producers in the United States and contains over 26 million acres of farmland [9]. The cornerstone of Arizona’s economy and culture, agriculture comprises four of the “five C’s of Arizona”—citrus, cotton, climate, and cattle (the fifth “C” being copper)—and generates over $23.3 billion annually [10,11]. Arizona is the United States’ second-largest producer of cantaloupe, broccoli, leafy greens, and dates [10]. Without a healthy workforce, the industry is unable to maintain this production level. As demonstrated in the meat processing plant industry [12], if production were halted due to disease spread and illness, significant economic losses likely would result [8].

To prevent human and economic impacts from SARS-CoV-2 outbreaks in the agriculture sector in Arizona, the University of Arizona Yuma Center of Excellence for Desert Agriculture (YCEDA)—in coordination with Yuma County Public Health Services District (YCPHSD), the Regional Center for Border Health, Inc., and the Arizona Department of Health Services (ADHS)—launched a wastewater-based surveillance (WBE) system to monitor SARS-CoV-2 RNA for public health response across public and private sectors, including agriculture. This partnership was formed to maintain the wellbeing of the worker population, primarily in addition to the food processing facility. Although the study is not the first to report building-level sampling or performing WBE for public health benefit that either promotes subsequent clinical interventions and educational practices for infection control [13,14,15], this study is unique in describing its applicability in a food-processing plant for the purpose to protect the employees and prevent unwanted food system closures. The objective of this manuscript is to highlight several case studies within a single food processing facility in Yuma, Arizona, which handles and packages food. In these case studies, WBE led to clinical testing, furloughing of positive workers, and containment of the infectious disease, demonstrating the utility of this monitoring approach.

## 2. Materials and Methods

### 2.1. Study Design

Between November 2020 and March 2022, one food processor facility’s wastewater was monitored for SARS-CoV-2 RNA. The facility is located in Arizona, approximately 30 miles from the United States-Mexico border, a company with annual revenue of over $30 million and customers in over 30 countries worldwide. During the study period, the monitored facility employed 277 workers, of which 246 were food handlers, 27 were administrative, and four were human resources professionals. Most workers were White, Hispanic (99%), and male (57%), with a mean age of 44 years old. Employees worked 8-h shifts, donned gloves when handling food products, and were required to follow public health guidelines (i.e., don masks and face shields) on-site.

### 2.2. Wastewater Collection, Processing, and Analysis

Wastewater was collected at the in-flow of a concrete septic tank (~11,500 L), fed solely by the facility’s plumbing. At least once every three-to-four days, a 1-L sample of raw sewage was collected from the septic tank. Samples represented 10-h composites since the septic tank pumped and flushed after reaching capacity, approximately every 10 business hours. The septic tank was also flushed after each event where individuals were diagnosed with SARS-CoV-2 on-site. SARS-CoV-2 particles were vacuum filtrated and concentrated via Centricon Plus-70 centrifugal filters, nucleic acids were extracted, and the N1 gene was enumerated via reverse transcriptase and quantitative PCR (RT-qPCR). Each collected sample was analyzed in triplicate, and the determination of a positive wastewater sample was based on the mean PCR quantification cycle. We spiked a subset of community wastewater samples (*n* = 41) with intact human coronavirus 229E. The mean recovery efficiency rates ranged between 6.81% to 31.60%. A standard curve (10^0^–10^7^ GC/L) for the N1 assay characterized RT-qPCR performance, including the slope (−3.39), y-intercept (41.37), and efficiency (97.4%). The limit of detection (2.00 × 10^3^) and the limit of quantification (3.60 × 10^3^) were reported per liter. The methodology is further detailed in a previous manuscript published by this group [14].

Although the ideal interpretation would estimate absolute case numbers via calculations using the concentration of RNA in a given volume of wastewater, the individual waste-shedding rate still has not been fully elucidated [16]; therefore, a binary interpretation—substantial detection (positive) or no substantial detection (negative)—guided public health decision-making. Because false negative tests could lead to high economic and potentially severe human health implications among the facility’s employees, the threshold to categorize positive samples was designed to be highly sensitive for this population. For example, in clinical studies, positive samples have typically corresponded with quantification cycle (C_q_) < 37, while C_q_ values from 37–40 were considered indeterminate [17]. Alternatively, in this study, the wastewater sample was interpreted as positive only if the N1 nucleocapsid region were detectable at C_q_ < 40; this resulted in several samples being interpreted as ‘positive’ despite being below the estimated limit of detection (LoD). All results—both positive and negative—were communicated with the food processing facility.

### 2.3. Public Health Response

Upon finalized reports, positive samples triggered an immediate public health response that consisted of auditing and reinforcement of infection control protocols, including hand hygiene, masking, social distancing, and physical barrier implementation in communal environments (e.g., the cafeteria). Similarly, facility leadership heightened awareness of disease prevalence among employees during morning and afternoon huddles to reinforce infection control protocols until wastewater samples were negative via RT-PCR. While follow-up clinical testing and vaccination services could not always be provided due to limited resources, in several cases, the facility implemented a point prevalence survey for SARS-CoV-2 infection via rapid antigen testing in addition to offering the Moderna SARS-CoV-2 vaccine for eligible employees. Three COVID-19 antigen tests were deployed based on accessibility: Abbott BinaxNOW, BD Veritor, and Care Start. However, individuals who administered tests did not document in what quantities each test was used. During these five events, all agricultural workers were tested via nares swab samples collected within 24 h of the positive wastewater result. Positive individuals and their close contacts were furloughed until deemed safe to return to work, in compliance with the contemporary United States Centers for Disease Control and Prevention (CDC) guidance. After the identification and isolation of positive individuals through clinical testing, wastewater samples were collected and tested within 24–72 h to determine if any individuals had residual viral shedding. If wastewater tested positive for SARS-CoV-2 RNA in subsequent collections, employees and wastewater were tested again, and so on; the entire surveillance and response structure is illustrated in Figure 1. It is important to note that clinical testing was entirely voluntary, and over 95% of employees chose to participate in testing during each event; the employees expressed a desire and comradery to protect each other from infection.

### 2.4. Contextualization of SARS-CoV-2 Detections

From November 2020 onward, wastewater samples were collected consistently on a biweekly basis, except during the facility’s off-season, between 1 July 2021 and 2 August 2021, when the food processor facility suspends operations annually. During the study period, the Wild-type and variants of the SARS-CoV-2 virus circulated in the surrounding region. Therefore, we have contextualized wastewater and clinical testing results within the larger scope of the predominant SARS-CoV-2 variants experienced by the United States population, using two CDC MMWR manuscripts [18,19,20]; this is visualized in Figure 2, along with the reported COVID-19 case counts in the food processor facility’s surrounding community, Yuma, Arizona (data provided by USAFacts).

## 3. Results

Wastewater was positive for SARS-CoV-2 RNA on 18 distinct sampling days (Figure 2) from a total of 205 collected and analyzed samples (8.8%), seven (38.9%) during the Wild-type predominance (from November 2020 to April 2021), zero during Alpha- and Delta-variant predominance (from April 2021 to July 2021 and July 2021 to November 2021, respectively), and 11 (61.1%) during Omicron-variant predominance (from November 2021 to March 2022) (the period is henceforth denoted by the variant name). In five of those events (two during Wild-Type and three during Omicron), on-site employees underwent rapid antigen testing the following day to identify SARS-CoV-2 infected individuals and furlough them to prevent transmission. During the subsequent days after testing positive for COVID-19, no workers self-identified as becoming symptomatic.

The morning of Wednesday, 2 December 2020, marked the first positive wastewater sample (C_q_ = 35.66; N1 = 3.9 log_10_ GC/L). On the following day, all 191 individuals who worked on-site in the previous week underwent rapid antigen testing, among whom four individuals tested positive (2% prevalence) and were all asymptomatic. The four positive workers and one positive worker’s spouse—also employed by the food processing facility and tested negative—were furloughed the same day. The wastewater basin was flushed later that day. On the next business day, 7 December, the wastewater was tested again, and no SARS-CoV-2 RNA was detected. No workers reported symptoms during the following days after the removal of the first four individuals from the population. The negative wastewater samples suggested that all infected employees had been identified and removed from working in the facility.

In the second instance when clinical testing was performed, SARS-CoV-2 RNA concentrations above preset threshold levels were detected on 22 February 2021 (C_q_ = 38.99; N1 = 2.7 log_10_ GC/L,). All workers who were staffed on-site that week (*n* = 186) participated in antigen testing the same day. One agricultural worker tested positive for SARS-CoV-2 and was furloughed; this individual was asymptomatic. The wastewater basin was flushed the same day. On the next business day, wastewater was collected again and tested negative for SARS-CoV-2 viral RNA (Figure 2).

On five other occasions during Wild-Type predominance, SARS-CoV-2 RNA was detected in wastewater: 21 December 2020; 5 January 2021; 7 January 2021; 14 January 2021; and 19 January 2021. In lieu of individual testing, facility leadership re-educated and audited compliance with infection control practices, including hand hygiene, proper masking, and social distancing. These practices alone, without the use of clinical testing and furloughing of positive workers, may have resulted in five out of six wastewater samples testing positive between 21 December 2020 and 19 January 2021. Given that fecal shedding can last between one to 33 days [21,22,23], the number of individuals who were positive during the 29-day period is unknown. Regardless, after January 19, no additional positive samples were identified throughout the next several weeks of wastewater analysis.

During Omicron predominance, SARS-CoV-2 RNA was detected in wastewater on 11 distinct sampling dates. The date of 5 January 2022 marked one of the highest concentrations of N1 measured in wastewater at the agriculture facility (C_q_ = 31.56; N1 = 5.2 log_10_ GC/L). This occurrence prompted a vaccination and testing campaign on 6 January 2022. Vaccine booster shots were administered to 117 workers, and nine workers were identified as positive for COVID-19 via antigen testing (8% prevalence). Positive individuals were furloughed immediately. Although RNA concentration declined on the next sample dates), N1 was still detected with high concentrations (C_q_ = 37.31; N1 = 4.1 log_10_ GC/L on January 13 and C_q_ = 36.82; N1 = 4.3 log_10_ GC/L on January 18. This incentivized a subsequent clinical testing campaign on 20 January 2022. Eighteen individuals were positive via antigen testing, which led to furloughing of those individuals in addition to two spouses who were also the facility’s employees (but tested negative). Subsequently, high RNA concentrations were detected on 1 February 2022 (C_q_ = 32.45; N1 = 5.3 log_10_ GC/L), prompting additional testing of 225 workers on February 3, among whom three were positive (1% prevalence). One additional worker who was not tested on-site reported a positive test result the same day. Individuals were furloughed, and upon resampling of wastewater, RNA concentrations dramatically decreased to levels considered negative (C_q_ = 40.65) the next day. During subsequent sampling on 7 February 2022, SARS-CoV-2 RNA was undetected (Figure 2).

## 4. Discussion

WBE efforts detected positive SARS-CoV-2 signals for 18 days at a food processing facility near the United States-Mexico border from November 2020 to March 2022. Detections occurred during Arizona’s second and third COVID-19 infection waves [24] (Figure 2). YCEDA communicated both negative and positive (Cq < 40 for SARS-CoV-2 N1 gene) results to the food processing facility leadership. In collaboration, both partners discussed public health measures to prevent infection spread. Regardless of the stockpile availability and the facility’s capacity to conduct clinical antigen testing, positive results triggered reinforcement of infection control activities, including reinforcement of correct face covering procedures and social distancing when feasible. In addition to good infection control practices, leadership reiterated positive results to employees during shift huddles. Unfortunately, we could not directly evaluate the impact of infection control interventions alone to prevent transmission without clinical testing to identify cases. Similarly, without the removal of COVID-19-positive workers, individuals may have continued to shed SARS-CoV-2 RNA in the septic tank, leading to several weeks of positive samples. Therefore, for up to 33 days of sustained wastewater positivity [25], we were unable to determine if the transmission had continued or if only prevalent cases were detected. Increased infection control practices alone may not have been sufficient to prevent transmission, which highlights the utility of pairing wastewater testing with clinical testing.

Conversely, negative wastewater results may not have represented actual SARS-CoV-2 infections, or lack thereof, in the facility. For example, the non-detect on 11 January 2021, where juxtaposed collection dates had positive samples, may have been caused by a lack of shedding from infected individuals or that those positive individuals did not defecate in the facility’s restrooms.

In at least three cases, two during the Wild-Type and one during the Omicron, positive wastewater samples led to possible breaks in transmission and, therefore, potential outbreak aversions. In those three events, after wastewater was determined to be positive for SARS-CoV-2 RNA, and infection control guidelines were reinforced, clinical testing was performed: all employees were tested, and positive workers were identified and furloughed, thereby removing potential sources for onward transmission from those individuals. Interestingly, the utility of test-and-furloughing yielded different results based on the period of the COVID-19 pandemic. WBE and clinical testing diagnostic pairings were perhaps the most impactful during Wild-Type predominance when employee furloughing led to an immediate reduction in SARS-CoV-2 RNA in wastewater to undetectable levels. This observation may be due to the relaxation of early, stringent county-wide policies during the later phases of the pandemic, including mask-mandate policies [26].

No employees reported illness, nor was wastewater positive during Alpha or Delta, which may be due to low case prevalence in the surrounding area (Figure 2). However, during Omicron, concordance between the removal of employees identified via antigen testing rarely led to an immediate decrease in SARS-CoV-2 RNA concentrations in subsequent collections. Unfortunately, the low sensitivity of antigen testing to identify individuals with early viral shedding was shown through preliminary research [27] and, evidenced by the activities that occurred from 1–4 February 2022, may have contributed to false-negative WBE results. Similarly, the specific antigen testing kits that were used and in what numbers were undocumented, resulting in unknown potential false-result rates. Regardless, the removal of positive individuals—who were determined negative the previous day via antigen testing—led to almost undetectable SARS-CoV-2 concentrations in the wastewater on the February 4 collection date.

Both during Wild-Type and Omicron predominance, SARS-CoV-2 RNA concentrations in the facility’s wastewater peaked and crested. However, our interpretations regarding transmission events shifted relative to the variant predominance. During Wild-Type predominance, when COVID-19 was novel and the scientific information regarding the virus was rapidly expanding, individuals adhered to masking practices and social distancing more rigorously both at work and in the community. Therefore, the transmission was more likely to have occurred during necessary congregation on shift. However, during Omicron predominance, “worry fatigue,” and COVID burnout were apparent [28], and although individuals adhered to infection control practices when actively at work, many individuals were reported to interact without social distancing or masks in parking lots and in the community. This change in behavior may have increased the likelihood of transmission in the community rather than on shift.

Several limitations are evident in this work which may compromise findings from both clinical and wastewater results. Regarding clinical testing, several documentation flaws were noted, including lost documentation regarding the exact number of individuals who were tested on January 6 and which specific COVID-19 antigen tests were used. Therefore, the individual diagnostic capabilities of each test may have biased diagnostic results [29]. Regarding wastewater testing, false negative samples may have occurred, dependent upon workers’ bathroom behavior. If individuals did not defecate in the facility’s bathrooms, their viral shedding status would not be represented in the wastewater samples. Therefore, it is possible that positive shedders may have been missed, which would then not trigger an immediate public health response. As such, some individuals may call into question the utility of wastewater-based epidemiology as opposed to consistent antigen testing twice or thrice weekly. The latter approach would be infeasible due to the resource intensity to coordinate the deployment of antigen test kits in addition to the likely intolerance of conducting nasal swabs so regularly, which has been documented to elicit significant discomfort.

As the pandemic progressed, we gained more familiarity and knowledge regarding how to use wastewater as a tool to institute public health action within the facility. During Wild-Type predominance, acquiring and deploying COVID-19 antigen tests was challenging, resulting in multiple instances where wastewater was positive with a lack of follow-up clinical testing (Figure 2). During Omicron predominance, because multiple community exposures were more plausible than in earlier COVID-19 phases, the test and furloughing method which was used prior may have been thought by the group to not be as effective. However, when coordination aligned and when exposures outside of working hours were unlikely, both the facility employees and the facility clearly benefited from the wastewater-based epidemiology and public health measures employed. Similarly, the partners formed stronger ties throughout time, which led to quicker and more seamless coordination.

The food processing facility never suspended production during the monitored period, whereas neighboring private businesses and schools were forced to shut down until outbreaks could be controlled—one business even closed permanently due to a COVID-19-related death. Consequently, this food processing facility could maintain the workforce and continue employing the hundreds of people who relied on this income while also serving the dozens of countries that purchased the company’s dates. The food processing facility attributes its success to several factors. First, a strong sense of community is shared among the facility’s staff. Staff refer to their coworkers as “a second family,” and many of the facility’s employees are related either biologically or through nuptials. This community perspective encouraged facility leadership to pursue WBE collaborations with YCEDA initially: to monitor the situation with real numbers and prevent their members from preventable illness. Workers were encouraged to contribute to the wastewater for accurate correlations between wastewater sample analysis and the presence of infection, and all staff understood the importance of testing and vaccination to prevent the spread of illness: the facility reports a >90% vaccination rate, at least 15% higher than Arizona’s ~74% vaccination rate [30]. For the first 15 people who were identified via clinical testing, the company provided wages until they could be cleared based on CDC guidance. After 15 people were identified throughout the pandemic, workers were requested to use their allotted paid leave until they were cleared to return. Furloughed workers received regular wages until they were cleared to return to work. Additionally, the food processing facility, YCEDA, and the Center for Border Health, Inc. communicated wastewater results regularly, assisted in interpretation and decision-making, and coordinated the deployment of vaccine and testing resources. Plans to automate communications efforts are in development, including the construction of a real-time dashboard, which sends email alerts to participating entities when updated. During these efforts, all institutions regarded ethical involvement, community empowerment, and worker safety as our predominant objectives, uniquely qualifying this as a community-based participatory research study [31].

This work is also novel through this intensive public-private partnership. Several studies have demonstrated the efficacy of WBE in monitoring and preventing COVID-19 outbreaks in aircraft [32], cruise ships [32], and college dormitories [14,33], emulating essentially closed systems, and the impact of largescale WBE use in a large urban setting [34]. Similarly, the CDC has initiated the National Wastewater Surveillance System to capture municipal wastewater data across the United States to track cases of the ongoing SARS-CoV-2 pandemic and intervene when necessary. However, a longitudinal implementation of WBE that focuses on a single private industry entity has not yet been detailed and reported in the literature. This series of cases from a food processing facility generalizes the value of WBE in several ways. Perhaps most notable, this study demonstrates the power of community and collaboration in public health, both within the food processing company, which professes a sense of familiar responsibility for the health of all employees, and the private-public partnership among the three participating organizations whose mission was to provide rapid testing and communication, as well as discuss practical implementation of public health interventions. Secondly, unlike the college dormitory studies, which studied a primarily White, non-Hispanic population with a narrow age range [14], this occupational setting was comprised of a relatively older Hispanic population. Further, this study suggests the benefit of applying paired WBE and clinical testing methods to an economically driven setting that can mitigate economic loss through worker illness and absenteeism.

This study has possible implications for border security between the United States and neighboring countries. Understanding the epidemiology of COVID-19 among individuals who reside in one country and work in a neighboring country may provide critical information for early warning, preparedness, and heightened awareness for possible surges in case numbers, especially during times when testing capacity may be burdened. Similarly, whole-genome sequence technologies may provide early warning to novel variant introductions. Such information should never be used to ostracize or scapegoat individuals and should be used solely for public health and disease prevention efforts for all individuals.

In conclusion, WBE has applications in diverse settings, exemplified by this case study in a food processing facility near the Mexico-United States border. Wastewater monitoring for SARS-CoV-2 informed public health action overall—including infection control response and clinical testing. In combination with clinical testing, WBE potentially mitigated two outbreaks, demonstrating that when adopted by public health and private industry, WBE may prevent widespread illness and, thus, economic losses.

## Figures and Tables

**Figure 1 viruses-14-02684-f001:**
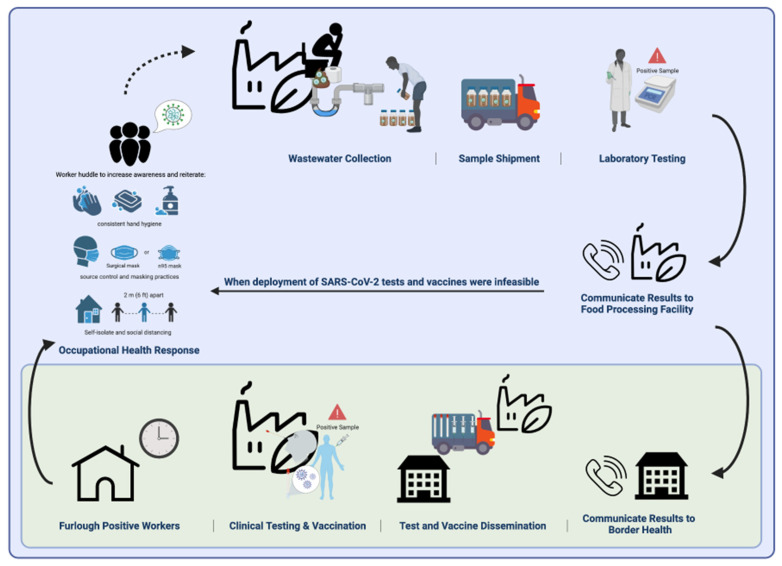
Wastewater Based Epidemiology Process to Monitor SARS-CoV-2 and Conduct Public Health Action an Arizona Food Processing Facility. Wastewater samples were collected biweekly from a food processing plant’s septic tank, shipped to a laboratory, processed, filtered, and analyzed for SARS-CoV-2 N1 RNA). Regardless of the feasibility to deploy SARS-CoV-2 antigen tests and vaccines, positive and negative samples were communicated with the food processing facility’s point of contact; all positive signals were communicated with food processing facility leadership and public health. Facility leadership increased awareness among staff and reiterated important infection control protocols to prevent further transmission within the facility. When deployment of tests and vaccines were feasible, results were communicated to Border Health representatives, who deployed medical products. Individuals diagnosed with SARS-CoV-2 via antigen tests were furloughed until deemed safe to return to work, in compliance with CDC guidance. Adapted from “COVID-19 Safety Information”, by BioRender.com (2022). Retrieved from https://app.biorender.com/biorender-templates. Images also provided by https://www.svgrepo.com

**Figure 2 viruses-14-02684-f002:**
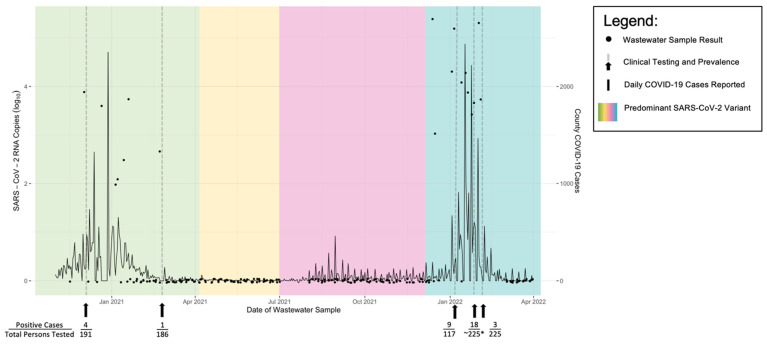
Wastewater Tested for SARS-CoV-2 N1 RNA in an Arizona Food Processing Facility, Nov 2020-Mar 2022. Wastewater samples were collected from a food processing facility’s waste basin and tested for SARS-CoV-2 N1 RNA between November 2020 and March 2022 (*n* = 205). During the testing period, 18 unique dates had positive detections for SARS-CoV-2. All results were communicated with facility management and in the event of a positive signal, public health response was conducted the agriculture processing plant, among border health, and public health personnel. Signals which led to clinical testing (*n* = 5) consistently identified SARS-CoV-2 positive employees. Dotted lines imbedded in the figure and arrows located below the figure indicate clinical antigen testing with the associated employees tested and number of positive detections below. COVID-19 cases reported to Yuma, Arizona are illustrated as a line graph to provide context regarding the COVID-19 activity in the surrounding community (data provided by USAFacts). Estimated predominantly circulating SARS-CoV-2 strain are reflected by background color panels: green, Wild-type; yellow, Alpha; pink, Delta; blue, Omicron. * Documentation of total individuals tested on 6 January 2022 was compromised and the denominator is an approximation.

## Data Availability

Most data used in this analysis are publicly available or available upon request to the corresponding author.

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
