# Peer review of "Wastewater-Based Epidemiology Mitigates COVID-19 Outbreaks at a Food Processing Facility near the Mexico-U.S. Border—November 2020–March 2022"

_viruses, 2022, doi:10.3390/v14122684_

Round 1

Reviewer 1 Report

This study demonstrates how wastewater surveillance in institutions can help identify asymptomatic cases to prevent further spread of COVID-19. During this COVID-pandemic, the findings are significant  and I agree that it should be published in this journal. However, I would like to the authors to address the following comments:

Line 98-99. I agree to avoid redundancy the authors have cited previous study for the methodology. But i would strongly request the authors to be kind to the viewers and at least add the recovery of enveloped virus in that line. The authors also did not mention the standard curves of the qPCR runs for this study. Were the samples of cited paper and the samples of this study tested in the same qPCR run? Either way please include.

The ethical clearance from an IRB was taken but details about antigen testing is not mentioned in methodology. If commercial kits were used, it should be clearly mentioned as different kits can have different sensitivity and specificity.

Line 271. deploy antigen deployments? please rephrase.

Author Response

The authors offer great thanks to the reviewers who took the time to add revisions and good suggestions. Please see the attachment for specific responses. Thank you

Reviewer 2 Report

Review for Viruses

Manuscript: viruses-2033307

Manuscript title: Wastewater-based Epidemiology Mitigates COVID-191 Outbreaks at a Food Processing Facility near the Mexico-2 U.S. Border—November 2020–March 2022

Authors: Innes et al

General comments

The MS describes the implementation of WBE to a food processing facility in Arizon. The focus of the WBE approach was to keep the facility running without ill workers or absenteeism. The focus of the MS was to describe the complex partnership and communication between analyzing laboratory, facility and official pandemic management along the ~200 days of monitoring. Moreover, some details on how wastewater signal was interpreted and how partners reacted to information was described.

The MS is very confusing. The topic is complex because reactions to wastewater signal changed over time. The MS does not present a structured and transparent approach on how to handle the implementation of WBE in such environments. It is a series of case reports during an evolving understanding of the pandemic and the WBE-approach.

Major concerns

·       The problem with monitoring facilities is, that we cannot be sure, that workers defecate at work. Hence, probability is high to generate a false negative signal from wastewater. Which strategy does the authors have to compete this problem?

·       This problem increases, when non-adequate sampling-techniques are used. Grab samples from a septic tank may not be representative for a community connected to this tank. Further information on recirculation of wastewater and flushing-intervalls of the tank are needed.

Minor concerns

·       For a practical implementation, the effort for data interpretation and communication was very high. It would be interesting how interpretation and communication can be automatized? What are the possibilities to streamline the communication flow and the resulting actions?

·       According to Fig 2, a figure including the communication processes (information flow) as well as a figure containing the measures for each event may make the complex topic more transparent.

·       A statement on how practical and how (personal) resource intensive was the WBE approach is missing. For example: would it be cheaper or easier to test every worker three times per week via antigen-test?

Author Response

The authors offer great thanks to the reviewer who took the time to add revisions and good suggestions. Please see the attachment for specific responses, and hope that these responses satisfy the reviewers' requests. Thank you

Round 2

Reviewer 2 Report

No further comments or suggestions for authors.